# Tongue of the Egyptian Endemic Bridled Skink (*Heremites vittatus*; Olivier, 1804): Gross, Electron Microscopy, Histochemistry, and Immunohistochemical Analysis

**DOI:** 10.3390/ani13213336

**Published:** 2023-10-26

**Authors:** Ramadan M. Kandyel, Om Prakash Choudhary, Sahar H. El-Nagar, Donald B. Miles, Mohamed Abumandour

**Affiliations:** 1Department of Zoology, Faculty of Science, Tanta University, Tanta 31527, Egypt; 2Department of Biology, Faculty of Arts and Sciences, Najran University, Najran 61441, Saudi Arabia; 3Department of Veterinary Anatomy, College of Veterinary Science, Guru Angad Dev Veterinary and Animal Sciences University (GADVASU), Rampura Phul, Bathinda 151103, Punjab, India; 4Department of Animal Wealth Development, Faculty of Veterinary Medicine, Kafrelsheikh University, Kafr El-Sheikh 33516, Egypt; 5Department of Biological Sciences, Ohio Center for Ecological and Evolutionary Studies, Ohio University, Athens, OH 45701, USA; 6Department of Anatomy and Embryology, Faculty of Veterinary Medicine, Alexandria University, Alexandria 21321, Egypt

**Keywords:** lingual papillae, laryngeal mound, immunohistochemical, SEM, *Heremites vittatus*

## Abstract

**Simple Summary:**

The present study describes the morphological characteristics of the tongue and laryngeal mound of *Heremites vittatus.* In SEM analysis, the complicated papillary system showed a special filiform system (nine subtypes) in addition to the conical system on the ventral surface of the foretongue apex. The dorsal surface of the laryngeal mound represented 18 longitudinal folds. Histologically, the keratinized layer was present in the foretongue but was absent in the mid- and hindtongue. Significant histochemical signals with positive glandular AB and PAS-positive reactions were observed. Immunohistochemistry showed strong cytokeratin immunopositivity in all parts of the tongue.

**Abstract:**

The present study used light and scanning electron microscopy to describe the integrative morphological description of the tongue and laryngeal mound of *Heremites vittatus*, an endemic lizard of Saharan Africa. Additionally, ultrastructure, histology, histochemistry, and immunohistochemical approaches were used to characterize the lingual apparatus adaptations. In the present study, *Heremites vittatus* consisted of a complex lingual papillary system in which the ventral apical surface of the foretongue comprised conical papillae. The dorsal surface consisted of different filiform papillary (*papillae filiformes*) types: the anterior section had two types (bifid and pointed), and the posterior section had four types (triangular, trifid, quadrifid, and pentafid) papillae. The dorsal midtongue surface exhibits scale-like, serrated filiform papillae with anterior gland openings. The hindtongue consisted of two overlapping filiform papillae: scale-like, board-serrated papillae on the median portion and finger-like papillae on the wings. The dorsal surface of the laryngeal mound had 18 longitudinal folds with glandular openings. Histologically, the foretongue was covered by a slightly keratinized layer that was absent in the mid- and hindtongue. The lingual glands were absent from the foretongue but present in the interpapillary space in the mid- and hindtongues. We observed a few rounded taste buds in the conical papilla epithelium. Histochemical analysis revealed strong glandular Alcian Blue (AB)-positive and Periodic Acid–Schiff (PAS)-positive reactions. Immunohistochemistry showed strong cytokeratin immunopositivity in all parts of the tongue. In conclusion, the obtained data about the lingual characterizations have been consistent with the active foraging behavior of the species and its environmental conditions.

## 1. Introduction

The feeding strategy of a species is a core behavior that significantly influences survival and reproductive success [1,2]. The tongue is considered a key innovation in the evolution of the terrestrial lifestyle as it is involved in prey processing and the passage of food from the oral cavity to the gut [3,4]. The tongue serves various functions, including prey capture, drinking, respiration, and protective behavior [5,6].

Reptile tongues are distinguishable from other vertebrates by the morphological and physiological attributes of individual species that correlate with feeding behavior (e.g., prey capture, prey processing) and lingual functions [3,7,8], in which diet, nutritional habits, and food manipulation all play a role in the lingual surface variances between reptiles [9]. Prior research relied on light microscopy to describe the tongue surface structure of different reptile species [10,11], which revealed a wide range of morphological characteristics of the tongue, particularly lingual shape, papillary presence, and distribution. The presence or absence of the different papillary types on the dorsal lingual surface of the different reptilian species indicated the feeding behaviors and nature of the available food particles [12]. Meanwhile, the author has been aware of the similarities and variances in the lingual epithelial structure of many reptile species.

*Heremites* (=*Mabuya*) *vittatus* (*Bridled Mabuya* or *Bridled Skink*) is a skink species found in North Africa and the Middle East. *Heremites vittatus* is associated with open-sandy or stony soil with little grass or heavy vegetation. This species can commonly be observed near water, especially in the wetland region of Egypt. The diet of *Heremites vittatus* consists of insects [13].

There is scanty information regarding the microscopic features of the tongue and laryngeal mound of *Heremites vittatus*. Understanding more about the oropharyngeal cavity anatomy may aid in keeping these species from becoming endangered by describing the functional relationship of the oral cavity with their eating, predation, and respiration. Therefore, the primary purpose of our observational study is to characterize and provide complete gross and ultrastructure characterizations of the tongue with the laryngeal mound of *Heremites vittatus*. This study also uses different histological techniques (ordinary histological, histochemical, and immunohistochemical) to illustrate the lingual adaptations with the different feeding habits and available feeding particles.

## 2. Material and Methods

### 2.1. Sample Collection

Ten adults (six males and four females) of *Heremites vittatus* were captured alive in the Abo-Rawash district, Giza, Egypt. The collected reptile species were kept in the animal keeping facility at the Department of Zoology, Faculty of Science, Tanta University, Egypt. The housing of *Heremites vittatus* includes a reptile’s cage with a temperature range of 18–24 °C and 55–70% humidity; crickets, mealworms, cockroaches, earthworms, superworms, hornworms, and other prey animals can all be found in diets. All the lizards were put to death by decapitation after an anesthetic overdose with ketamine hydrochloride (Parke-Davis, Berlin, Germany), 325 μg/g body weight [14], carried out by a licensed zoologist in conformity with local, national, and international ethical standards, and examined for any anatomical anomalies. The samples that were being evaluated were brought to the lab’s facilities in ice boxes.

This study was carried out according to the Institutional Animal Care and Use Committee (IACUC) protocols of Laboratory Animals, Faculty of Science, Tanta University, and it was approved by Egyptian laws. The experiments were carried out following the ethical provisions established by the 2010/63/EU directive for animal experiments and were prepared according to the guidelines for the use and care of animals of the IACUC Tanta University, under research project number (IACUC-SCI-TU-0309).

### 2.2. Morphological Examinations

Heads were decapitated, and tongues and laryngeal entrances were carefully removed and submerged in 10% neutral buffered formal saline. After fixation, three tongues and laryngeal entrances were examined grossly using a stereomicroscope (Olympus VM VMF 2×, Eyepiece 10× Stereo Microscope, Tokyo, Japan). Then, the photographs of the tongue were taken with a digital camera a digital camera (Canon IXY 325, Tokyo, Japan). In this study, all the anatomical terminologies have been used according to Nomina Anatomia Veterinaria (NAV) [15].

### 2.3. Scanning Electron Microscopy (SEM) Analysis

The tongues of four individuals of *Heremites vittatus* were prepared for SEM techniques [16]. Each tongue with its laryngeal entrance was first rinsed in 2% glutaraldehyde (Mfcd00007025) dissolved in a 0.1 M phosphate buffer solution at pH 7.4 (Mfcd00131855). The tongues were then cut and fixed in 2% formaldehyde and 1.25% glutaraldehyde in a 0.1 M sodium cacodylate buffer (97068-100 ML-F), pH 7.2 at 4 °C. Samples were washed in 0.1 M sodium cacodylate containing 5% sucrose, processed by tannic acid (Mfcd00066397), dehydrated in ascending grades of ethanol (50%, 70, 80, 90, 95, and 100% ethanol, 15 min each), then held and processed for 2 h at room temperature [16]. Finally, the samples were dried in carbon dioxide (critical point drying), attached to stubs with colloidal carbon, and coated with gold–palladium in a sputtering device. SEM analysis was carried out with the JEOL JSM 6510 lv SEM unit at the Laboratory of Electron Microscopy at the Faculty of Agriculture, Mansoura University, Egypt. SEM image analysis was carried out at the Department of Veterinary Anatomy, College of Veterinary Science, Rampura Phul, Punjab, India.

### 2.4. Histological and Mucin Histochemical Examinations

Three tongue specimens (0.5 cm^3^) from the three different lingual regions (fore, mid, and hindtongue) of *Heremites vittatus* were collected and immersed in 10% neutral buffered formal saline for 24 h. The tongue samples were thoroughly transferred to 70% alcohol after 48 h. The tongue samples were dehydrated with increasing gradations of ethanol (70–100%), cleared in xylene, impregnated, and embedded in paraffin wax. Histological sections of 5–6 μm were cut using a Leica rotatory microtome (RM 20352035; Leica Microsystems, Wetzlar, Germany) and mounted on glass slides. Paraffin sections were organized and stained with Hematoxylin and Eosin Stain (AB245880) according to the Suvarna et al. [17] protocol. Extra-paraffin sections were stained with AB (pH 2.5, Mucin Stain) (ab150662) [18] and PAS (ab150680) [19] to localize the acidic and neutral mucopolysaccharides, respectively. Moreover, the collagen fibers were examined using Masson’s trichrome stain method (AB150686) [20]. The stained sections were examined with a BX50/BXFLA microscope (Olympus, Tokyo, Japan). From the micrographs, the height, width, and gland diameter of the papillae on different lingual parts were determined using ImageJ software (1.53t) [21].

### 2.5. Immunohistochemical Analysis of the Dorsal Mucosal Lingual Keratin

Lingual tissues were incubated overnight with the primary Anti-Pan Cytokeratin antibody [AE1/AE3] that was found as Invitrogen Anti-Pan Cytokeratin Monoclonal (AE1/AE3), eBioscience™, Catalog # 50-9003-82. After washing in a Phosphate-buffered saline (PBS), the sections were incubated for 30 min with Rabbit Anti-Mouse secondary antibody and then for 30 min with ABC, diluted to 1:200 [22]. After three PBS washes, immerse for 5–10 min in freshly prepared 0.1% 3, 3-diaminobenzidine tetrahydrochloride (DAB). Sections were counterstained in Mayer’s Haematoxylin for 2 min, dehydrated, cleared in xylene, and mounted. The primary antibody was replaced by PBS for the negative control test. Finally, sections were examined and photographed under the bright field light microscope (Olympus BX 50 compound microscope).

### 2.6. Negative Image Analysis Using CMEIAS Color Segmentation (Supplementary Image)

The negative image of Figure 8 was performed using CMEIAS color segmentation and improved computing technology. Firstly, the image was opened with CMEIAS Color Segmentation, then selected “Process” from the menu items and clicked “Negative Image” [23].

### 2.7. Digital Coloring of Scanning and Transmission Electron Microscopic Images

We digitally colored the scanned electron microscopy images using the Photo Filter 6.3.2 program to recognize the different lingual papillary regions, as per the method described by Kandyel et al. [24].

## 3. Results

### 3.1. Gross Morphological Observations of the Tongue and Laryngeal Entrance of the Heremites Vittatus

The tongue is triangular and broadens posteriorly, with a pointed, black-pigmented tip. The dorsoventral flattened tongue was classified into three lingual regions: the foretongue, the midtongue, and the hindtongue, with two lateral lingual wings encircling the laryngeal mound and its glottis (Figure 1/FT, MT, HT, LM). The average length of the tongue was measured at about 1.164 ± 0.13 cm. The three lingual regions had an average length of (0.270 ± 0.031, 0.392 ± 0.024, and 0.49 ± 0.41 cm) and an average width of (3.61 ± 0.15, 4.81 ± 0.74, and 5.61 ± 0.89 cm) of the foretongue, midtongue, and hindtongue, respectively. The ventral surface of the tongue was attached to the oral cavity floor by the lingual frenulum.

The pharyngeal cavity was occupied by the elongated laryngeal mound that carried a median longitudinal glottis opening. The laryngeal mound was situated directly posterior to the lingual root and surrounded by the two lateral lingual wings (Figure 1/LM, GO, LW). Moreover, our morphometric findings showed that the average total length of the laryngeal mound and glottic opening were 0.289 ± 0.032 and 0.120 ± 0.014 cm, respectively, while the average total width of the laryngeal mound and glottic opening were 1.93 ± 0.124 and 0.31 ± 0.012 cm, respectively. Additionally, the slight and non-significant increase in its equatorial diameter (0.208 ± 0.02 cm) was greater than the axial one (0.1038 ± 0.003 cm).

Our description depends mainly on the location of the lingual frenulum, as shown in (Figure 1) to describe the three distinct lingual parts (the foretongue, the midtongue, and the hindtongue), in which the foretongue begins from the rostral part of the lingual tip until the beginning of the lingual frenulum. The midtongue begins from the beginning of the lingual frenulum until the terminal portion of the lingual frenulum, and the hindtongue begins from the terminal portion of the lingual frenulum.

### 3.2. Scanning Electron Microscopy Observations of the Tongue and Laryngeal Entrance

The ventral surface of the foretongue’s apex has numerous conical papillae (*papillae conicae*) of hexagonal and pentagonal determinations with narrow space in-between (Figure 2/FT, VS, CP); additionally, with high magnifications, each conical papillae revealed the irregular ventral surface appearance (Figure 2E)**.** Whereas the dorsal surface of the anterior apical part had two types of posteriorly directed filiform papillae (*papillae filiformes*): bifid and pointed papillae with a corrugated surface (Figure 3A–E/FT, DS, FP, BFP, black star, OFP).

The dorsal surface of the posterior part of the foretongue possessed four types of posteriorly directed filiform papillae (*papillae filiformes*): the triangular, trifid, quadrifid, and pentafid papillae (Figure 3A,F/RFP, TFP, QFP, PFP, black star); additionally, with high magnification, there were a small number of taste buds on the dorsal papillary surface and a corrugated papillary surface at the basal part (Figure 3G–H/TB).

The ventral surface of the midtongue was attached ventrally to the oral cavity floor by the thick lingual frenulum (Figure 4A,B/MT, LF). The dorsal surface of the midtongue possessed numerous posteriorly directed scale-like serrated filiform papillae (Figure 4A–F/MT, SFP) that overlapped each other; additionally, with high SEM magnification, the dorsal papillary surface possessed numerous microridges, small openings of the anterior lingual salivary glands, and taste buds (Figure 4G–I/Mr, red arrowheads, TB).

The dorsal surface of the hindtongue had three portions: the median and the two lateral lingual wings (Figure 5A/HT and LW). Its filiform papillae were divided into two types that overlap each other: the scale-like board serrated posteriorly directed filiform papillae on the dorsal surface of the median portion of the lingual root (Figure 5A–C/SRFP) and the finger-like projected filiform papillae on the dorsal surface of the two lingual wings (Figure 5A,D,E/FFP), while the medioposteriorly directed finger-like projected filiform papillae were observed on the dorsal surface of the two lingual wings (Figure 5A,D,E/FFP). With high magnification, the dorsal surface of these papillae possessed a small opening of the posterior salivary glands on their dorsal surface (Figure 5C,F).

The laryngeal entrance region was formed by the laryngeal mound with its median glottic opening (Figure 5A,G/LM, GO). The dorsal surface of the laryngeal mound carried about 18 longitudinal folds that had a corrugated surface (Figure 5A,G,H/LM, FO); additionally, with high magnification, these folds carried numerous small openings of the laryngeal salivary glands (Figure 5I/FO, red arrowheads).

### 3.3. Histological, Histochemical, and Immunohistochemical Studies Examinations

Histologically, the numerous posteriorly directed lingual filiform papillae were dispersed along the dorsal surface with various morphological characteristics. The lingual papillary cores were formed by the connective tissue that was located above numerous muscle fiber bundles. These well-developed muscle bundles were organized into transverse and longitudinal courses (Figure 6A–C/CTC, MB)**.** The foretongue was covered by a stratified, slightly keratinized squamous epithelium that grew out to form the pointed filiform papilla (Figure 6A–C/PFP, KE). Deep interpapillary notches separated the papillae, and their cores were filled with the loose collagenous connective tissue of the submucosal layer (Figure 6A–C/PFP, IPS). There were no lingual glands or sensory structures observed.

The midtongue was covered by stratified non-keratinized squamous epithelium, which formed the pointed filiform papilla (Figure 6D–F/PFP, NK). Several alveolar-like lingual glands were identified at higher magnification in the interpapillary space regions (Figure 6E/LG). Furthermore, the papillae were supported by a layer of well-developed muscle fibers arranged in different patterns (Figure 6E,F/MB)**.** On the surface of the hindtongue, two different-shaped filiform papillae were detected; each one was covered by a stratified, non-keratinized squamous epithelium (Figure 6H–J/PFP, CFP, NK)**.** The pointed filiform papillae were distributed over the anterior part of the hindtongue, similar to those observed at the midtongue, while the conical filiform papillae covered the posterior part (Figure 6I,J/CFP). Lingual glands were situated in the interpapillary spaces and in the depth of the underlying muscular tissue (Figure 6H–J/LG). A few rounded-shaped taste buds were observed in the epithelium of the conical filiform papillae (Figure 6I/blue arrowhead).

The collagen fibers were intertwined and filled the core of each lingual papilla as well as between the muscle fiber bundles beneath (Figure 7/PFP, yellow arrowheads). The histochemical investigations revealed that the lingual glands displayed strong AB and PAS positive reactions, in which the blue color indicated positive AB reactivity, whereas the red one indicated PAS reactions (Figure 8 and Figure 9). Immunohistochemistry showed strong cytokeratin immunopositivity in all parts of the tongue (Figure 8, Figure 9 and Figure 10A,a,B,b).

Our statistical analysis of the height, width, and gland diameter of papillae on different parts of the tongue revealed that the papillae height and width significantly increased from the fore- to midtongue and non-significantly decreased towards the hindtongue. While the gland diameter gradually and significantly increases from the foretongue towards the hindtongue (Figure 11)**.**

## 4. Discussion

The present study revealed fascinating insights into how the examined lizards have adapted to survive and thrive in their Egyptian ecosystem. For example, some reptiles have developed specialized jaws and teeth to efficiently capture and consume their preferred prey, while others have adapted to survive on a diet that may include plants, insects, or even other reptiles [1]. Anatomically, these different feeding strategies influenced the lingual structure and its framework, and these appeared in the currently examined reptile species, *Heremites vittatus*. The studied species resembled most reptile species, and they depended on a prompt bite to capture their prey by using their jaws, whereas most lizard species use an actionable searching strategy [25].

Dietary specialization is thought to have two effects on lingual morphology, in which these adaptations were observed on the foretongue surface, which comes into direct contact with various food particles during feeding [26], and the lingual papillae, which aid in the movement of food particles towards the esophagus [12,25]. According to prior published data on the reptile species, the *Heremites vittatus* tongue was categorized into three regions: foretongue, midtongue, and hindtongue, with two lateral lingual wings that encircled the laryngeal mound and glottis [25,27,28]. Furthermore, the thick lingual frenulum attached the midtongue ventrally to the oral cavity floor, which was already described in most reptile species [25], including Bosc’s fringe-toed lizard and Sinai fan-fingered gecko. Also, Abbate et al. [29] reported that the blue-tongued lizard tongue was classified into three regions but with different nominations as follows: the lingual tip, foretongue, and hindtongue, while Cizek and Hamouzova [28] gave other lingual classifications of the sand lizard *L. agilis* as bifid apex, body, and bifurcated root. The present gross and SEM observations revealed that the tongue of *Heremites vittatus* has a triangle shape that represents the typical lingual shape in most reptilian species, similar to that described [25,27,30] in Bosc’s fringe-toed lizard *A. boskianus,* Sinai fan-fingered gecko *P. guttatus*, Leopard Gecko *E. macularius*, and Green Iguana *I. iguana*, respectively. Meanwhile, other lizards have unusual tongue shapes, such as the blue-tongued lizard’s V-shaped tongue [29].

The tongue has an essential biological role in all vertebrates, particularly in reptile species, as it is related to numerous functions. For example, the tongue of the *Agama stellio* has been used for hunting [6,25], and the tongues of the *Oplurus cuvieri* and *Iguana iguana* have been used for transportation and swallowing food particles [25,30,31]. According to the previously published data, numerous appearances of the lingual tip in different reptile species correlate with the available food particles [7,25]. The current work revealed the presence of a pointed, black-pigmented tip in the *Heremites vittatus* tongue. Meanwhile, the most common lingual tip appearance in most reptilian species is the bifurcated tip, as in *Anoliscaro linensis* [32], *Oplurus cuvieri* [31], *Psammophis sibilans* [7], the green iguana [30], Bosc’s fringe-toed lizard [25], and *Pogona vitticeps* [33], while a slightly bifurcated tip was observed in the tongues of the Sinai fan-fingered gecko [25] and *Gekko japonicus* [34]. Meanwhile, the round tip was reported in blue-tongue skinks [29]; additionally, the pointed or round tip in the current observations may assist in increasing the speed of protraction movement [7,25,29].

The current study verified previously published data indicating most reptile species used their projecting foretongue region to touch and taste the substrate, such as food or water [25] in Bosc’s fringe-toed lizard and Sinai fan-fingered gecko, while Cooper [35] indicated that tongue-flicking is the basic squamate behavior. Moreover, the current study confirmed that *Heremites vittatus* depended on the “actively hunting” feeding technique described in Bosc’s fringe-toed lizard [25], which predominantly utilizes chemical notification. In contrast, the Gecko species using a “sit-and-wait” feeding technique, similar to that described in the Sinai fan-fingered gecko [25], were thought to depend primarily on visual signals. In harmonization with Cooper [36] and proven by Baeckens et al. [37], who concluded that the lingual morphology and Jacobson’s organ were believed to mirror these features as well as the reptilian species utilizing chemical signals to communicate with hetero- or conspecifics, moreover, the described broad feature of the hindtongue region, similar to that observed in most reptilian species, Gewily and Mahmoud [25], may provide a wide surface accessible for loading water during drinking or bearing prey back to the oral cavity.

The lingual papillary system reveals distinct feeding strategies and lifestyles in various vertebrate species, including the currently examined *Heremites vittatus*, blue-tongue skink (*Tiliqua scincoides*), green iguana (*Iguana iguana*), Bosc’s fringe-toed lizard Acanthodactylus boskianus, Sinai fan-fingered gecko Ptyodactylus guttatus, Egyptian fruit bat (*Rousettus aegyptiacus*), and common quail (*Coturnix coturnix*) [8,25,26,29,30]. The unique “actively hunting” feeding strategies and lifestyles of *Heremites vittatus* contribute to their unique adaptations and survival in various environments. The examined *Heremites vittatus* tongue has a unique papillary system with distinct regional specifics with different directions, sizes, and distributions. They include a special filiform system on its dorsal surface and a conical system on the ventral surface of the foretongue region, which is suitable for its rapid prey capture using the “actively hunting” strategy. According to current SEM observations, the presence of papillary variations, including their sizes, shapes, directions, and distributions on the dorsal and ventral lingual surfaces, plays a mechanical role in the captured food handling; additionally, the interpapillary space acts as a lingual pouch to pick up a large quantity of the food particles contacting the lingual surface during its protrusion to gather airborne and substrate-fixed chemical particles, then retracts and transports them to the vomeronasal organ [7,38].

Our study on the *Heremites vittatus* tongue revealed a unique papillary distribution, with numerous conical papillae on the ventral surface, two types of posteriorly directed filiform papillae on the anterior foretongue’s dorsal surface, and four types of posteriorly directed filiform papillae on the posterior foretongue part. The midtongue region had numerous scale-like serrated filiform papillae, while the hindtongue region had scale-like board-serrated filiform. Finger-like projected filiform papillae were also observed on the dorsal surface of the two lingual wings. The description of the different papillary subtypes on the reptile tongue was recorded previously, in which the three papillary subtypes were observed on the dorsal surface of the Bosc’s fringe-toed lizard tongue, which named the conical flattened and conical round papillae on the fore- and midtongue and the long papillae on the hindtongue, while the Sinai fan-fingered gecko tongue consisted of the rounded cylindrical, conical papillae on the foretongue and midtongue regions, whereas the tall filiform papillae on the hindtongue [25]. Also, the three papillary subtypes with different nominations were seen on the dorsal lingual surface of the *Gekko japonicus* [34]: dome-shaped papillae on the apex, flat fan-shaped papillae on the body, and long, scale-like papillae on the lateral part of the apex and body. Iwasaki [34] reported that the gecko tongue had three papillary subtypes that were named: dome-like, flat leaf-like, and broad scale-like papillae on the fore-, mid-, and hindtongue, respectively.

Meanwhile, in lizards, there were broad scale-like papillae on the foretongue, transverse (parallel folds), broad short (folds), and leaf-like papillae on the midtongue, in addition to the presence of broad, flat, and fan-shaped papillae on the hindtongue [11]. Meanwhile, the two papillary subtypes with different nominations were reported by Abbate and Latella [29] in the blue-tongued lizard tongue, which named the foliate-like papillae on the median part of the dorsal surface of the fore- and hindtongue regions and the cylindrical papillae on the lateral part of the same regions, with the complete absence of these papillae from the lingual tip. In addition, Abbate and Latella [30] described the presence of the conical and cylindrical papillae on the dorsal surface of the fore- and hindtongue, respectively, with the complete absence of these papillae from the lingual tip of the green iguana. However, the reptile tongue in some turtles and snakes lacked lingual papillae [7]. In general, the absence of lingual papillae on the entire dorsal lingual surface of turtles [39] indicates that the presence of lingual papillae correlates with a species’ adaptation to its habitat, diet, or feeding behavior.

The current work revealed that the dorsal papillary surface of the *Heremites vittatus* tongue possessed a small number of taste pores on the foretongue and taste buds on the midtongue. The Sinai fan-fingered gecko tongue has numerous taste buds along its two-thirds lingual region, while the Bosc’s fringe-toed lizard tongue has a small number on its foretongue but an increase in its midtongue [25]. Previous studies [29,40] found that gekkotan and blue-tongued lizard tongues primarily lack taste buds, suggesting it may be a phylogenetic, functional, or adaptive feature. This view disagrees with that mentioned by Bayoumi and Abd-Elhameed [11], who observed taste buds on the apical and lateral surfaces of the dome-shaped papillae on the lingual tip of *P. guttatus,* while they were completely absent from the papillae of *A*. *boskianus* tongue; meanwhile, Taha [41] observed the presence of a few taste buds in the epithelium of the three lingual parts of *Trachylepis vittata*. Schwenk [5] supposed that in all iguanids, the taste buds were abundant on the papillary surface of the foretongue region and played an important role in testing the prey’s palatability when touching its tongue surface during the capturing process. The tongue of some lizards is utilized as a chemoreceptor organ to follow the pheromone trails of prey, as described in Bosc’s fringe-toed lizard [25].

The current histological study agrees with most published data that the dorsal lingual epithelium was formed from a stratified squamous type [11,25]. The previously published data showed that the common variations among the reptiles were focused on the presence, absence, or amount of keratinized material on the dorsal lingual surface, which is closely related to the feeding mechanism and the feeding habits of the different reptile species. The foretongue of *Heremites vittatus* is covered by a slightly keratinized layer that extends to the filiform papilla, whereas this keratinized layer completely disappears from the dorsal surface of the mid- and hindtongue. Meanwhile, Gewily and Mahmoud [25] described the presence of the keratinized layer over the dorsal surface of the midtongue of Bosc’s fringe-toed lizard and the ventral surface of the foretongue of both Bosc’s fringe-toed lizard and the Sinai fan-fingered gecko. However, the lingual keratinization was described only on the lingual tip of the scincine lizard [42], but the presence of the keratinization on the dorsal and ventral lingual surfaces was described in *Psammophis *sibilans,** with those on the dorsal surface reaching three times that present on the ventral surface [7]. In the white spot gecko, there is a thin keratinized layer on the dorsal surface of the foretongue region. In contrast, this keratinized layer is absent in the Egyptian spiny tail lizard, *U. aegypticus* [11]. Furthermore, the complete absence of keratinization from the dorsal lingual surface is a popular characteristic in some reptile species [4,10,27,43]. The *Heremites vittatus* tongue’s findings show that keratinization is absent from the mid- and hindtongue’s dorsal lingual surfaces, but it compensates with a complex papillary system on its dorsal and ventral lingual surfaces, containing numerous papillary types and microridges, and numerous conical papillae on the ventral surface of the foretongue region. Functionally, the numerous non-keratinized papillary types on the dorsal lingual surface may facilitate the slipping process of caught prey towards the esophagus. Moreover, it was described that the presence of microridges on the lingual surface helps in the spreading of the mucosal film (secreted by the lingual glands) over the lingual surface to reduce friction forces [44,45].

The lingual glands In the *Heremites vittatus* tongue were absent in the foretongue, whereas these glands were observed in the interpapillary space of the mid- and hindtongue. In addition, these lingual glands were observed in the depth of the underlying muscular tissue. The lingual glands were observed between the papillary bases and were numerous in the hindtongue [45,46] in adult *C. ocellatus* and *E. schneideri*, respectively. The lingual gland’s secretion acts as a lubricant material on the dorsal lingual surface to facilitate the caught food particle’s movements, transportation, and swallowing [25,46].

The current histochemical investigations of the examined *Heremites vittatus* tongue revealed that the collagen fibers were intertwined and filled the core of each lingual papilla and between the muscle fiber bundles beneath. The sheet of collagen connective tissue was observed just beneath the epithelial layer, as reported in *C. ocellatus* [45] and *E. schneideri* [46]. The obtained findings confirmed that muscle contraction will lead to shortening or curving of the papillae due to the presence of the collagenous sheet. Furthermore, the current findings agree with Winokur [39] that the lingual muscles attached to the lingual glands will help in squeezing the glands to secrete their secretion on the dorsal lingual surface. The current histochemical investigations of the *Heremites vittatus* tongue revealed that the lingual glands displayed strong positive reactions when stained with AB and PAS. The AB blue color indicates the acidic mucin, while the reddish–purple color refers to the neutral mucin secretions when stained with PAS stain. Meanwhile, in the *P. vitticeps* lizard [47], it was reported that the tubular serous lingual glands of the foretongue region gave PAS-negative secretory cells, but the tubulo-alveolar mucous lingual glands were PAS-positive, revealing the presence of mucopolysaccharide. Meanwhile, the lingual glands gave an AB-PAS-positive reaction [45] in adult *Chalcides ocellatus*.

The present findings represent the first data on the cytokeratin expression of the *Heremites vittatus* tongue. As expected, we observed that the cytokeratin expression showed strong immunopositivity in all parts of the tongue.

## 5. Conclusions

Conclusively, in *Heremites vittatus*, the complicated papillary system included filiform papillae on the dorsal surface (nine subtypes) as well as conical papillae on the ventral surface of the foretongue region, which was ideal for capturing prey quickly using the “actively hunting” method. Histologically, the foretongue was covered by a slightly keratinized layer that disappeared entirely from the mid- and hindtongue regions. The lingual glands exhibit strong positive AB and PAS reactions. Immunohistochemistry showed strong cytokeratin immunopositivity in all parts of the tongue.

## Figures and Tables

**Figure 1 animals-13-03336-f001:**
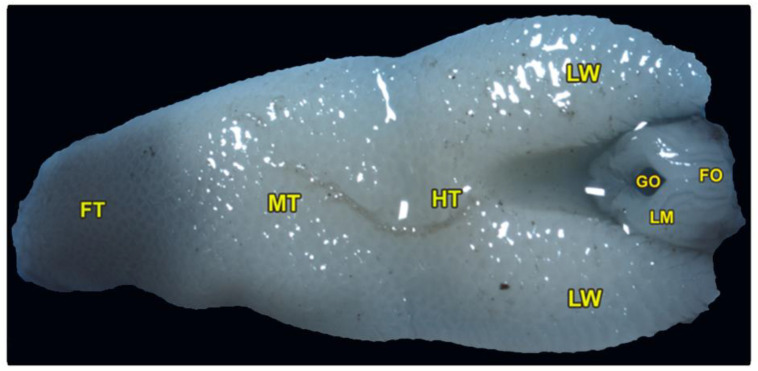
Gross morphology of the tongue of *Heremites vittatus*, showing the foretongue (FT), midtongue (MT), hindtongue (HT), the lingual wings (LW), the laryngeal mound (LM), the median glottic opening (GO), and laryngeal folds (FO).

**Figure 2 animals-13-03336-f002:**
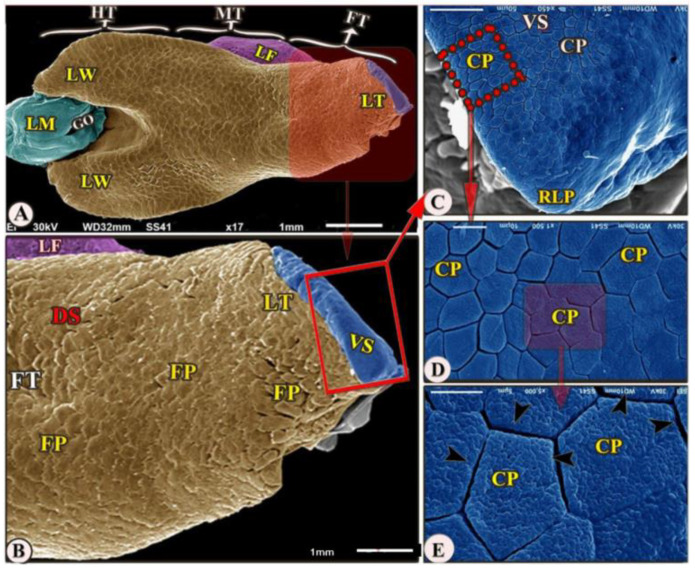
Scanning electron microscopic image (views (**A**–**E**)) of the *Heremites vittatus* tongue showing the foretongue (FT) with its lingual part (LT) with its anterior round tip (RLP), midtongue (MT) with lingual frenulum (LF), and hindtongue (HT). The dorsal surface (DS) possessed *papillae filiformes* (FP). The ventral surface (VS) possessed *papillae conicae* (CP) that separated narrow spaces (black arrowheads). The lingual wings (LW), the laryngeal mound (LM), the median glottic opening (GO) were clear.

**Figure 3 animals-13-03336-f003:**
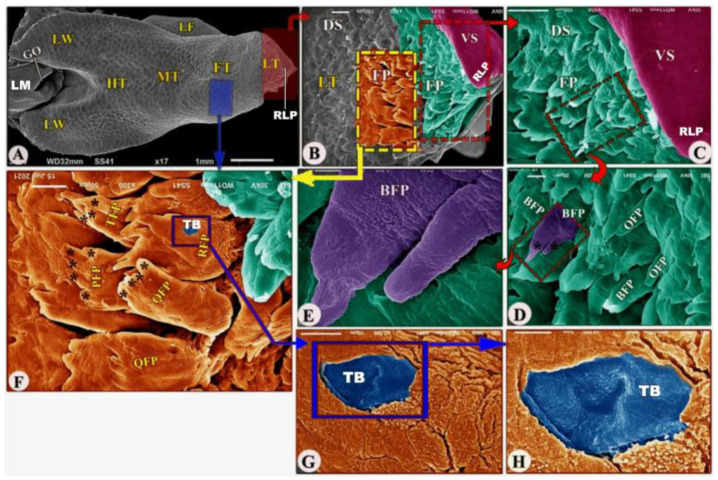
Scanning electron microscopic image (views (**A**–**H**)) of the *Heremites vittatus* tongue showing the foretongue (FT) with its lingual part (LT) with its anterior round tip (RLP), midtongue (MT) with lingual frenulum (LF), and hindtongue (HT) with its lingual wings (LW). The ventral (VS) and dorsal (DS) surfaces possessed filiform papillae (FP), bifid filiform papillae (BFP), pointed filiform papillae (OFP), trifid filiform papillae (TFP), quadrifid filiform papillae (QFP), pentafid filiform papillae (PFP) with processes (black star), triangular filiform papillae (RFP), and taste buds (TB). The laryngeal mound (LM) with the median glottic opening (GO) and laryngeal folds (FO).

**Figure 4 animals-13-03336-f004:**
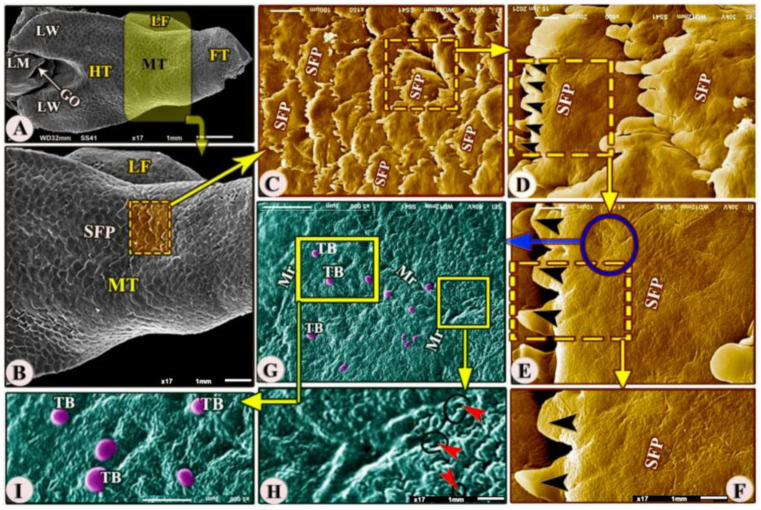
Scanning electron microscopic image (views (**A**–**I**) of the *Heremites vittatus* tongue showing the foretongue (FT), midtongue (MT), lingual frenulum (LF), and hindtongue (HT) with its wings (LW). The dorsal surface of the midtongue possessed overlapped scale-like serrated filiform papillae (SFP) with a serrated apex (black arrowheads). With a high magnification of the papillary surface, there were taste buds (TB), microridges (Mr), and anterior salivary gland openings (red arrowheads).

**Figure 5 animals-13-03336-f005:**
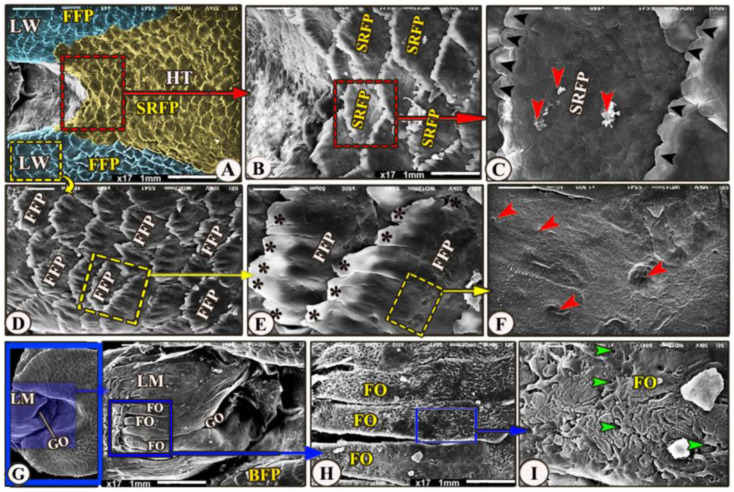
Scanning electron microscopic image of the *Heremites vittatus* tongue. (Views (**A**–**F**) show the dorsal surface of the hindtongue (HT) that carried scale-like board serrated papillae (SRFP) with a serrated apex (black arrowheads) in the median part, whereas the wings (LW) had finger-like projected papillae (FFP) with processes (black stars). The high magnification of the papillary surface showed numerous posterior salivary gland openings (red arrowheads). (Views (**G**–**I**)) showing the laryngeal mound (LM) and median glottic opening. (GO), with high magnification, its dorsal surface possessed 18 longitudinal laryngeal folds (FO) and numerous small openings of the laryngeal salivary glands (green arrowheads).

**Figure 6 animals-13-03336-f006:**
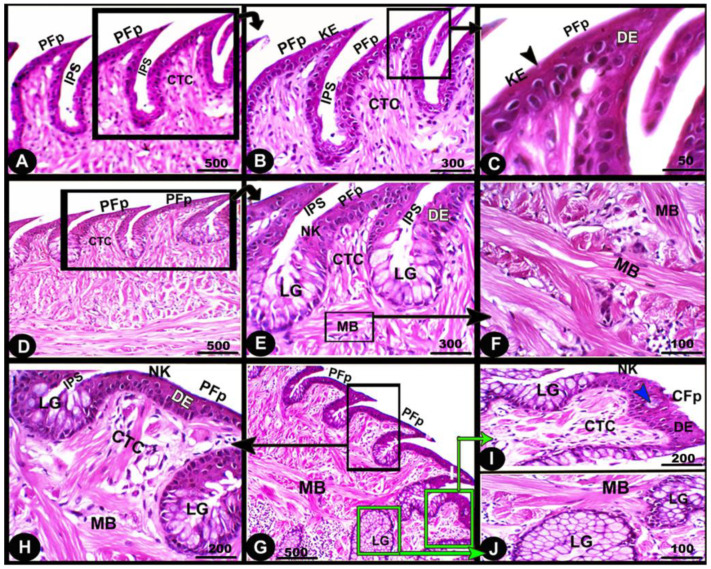
Micrograph image (views (**A**–**J**)) of the dorsal surface of the foretongue (views (**A**–**C**)), midtongue (views (**D**–**F**)), and hindtongue (views (**H**–**J**)) of the *Heremites vittatus* tongue. The pointed filiform papillae (PFP) and conical papillae (CFP), interpapillary space (IPS), connective tissue core (CTC), muscle bundles (MB), keratinized (KE) or non-keratinized (NK) dorsal squamous epithelium (DE), keratinized layer (black arrowhead), taste buds (blue arrowhead), and lingual glands (LG). H&E stain.

**Figure 7 animals-13-03336-f007:**
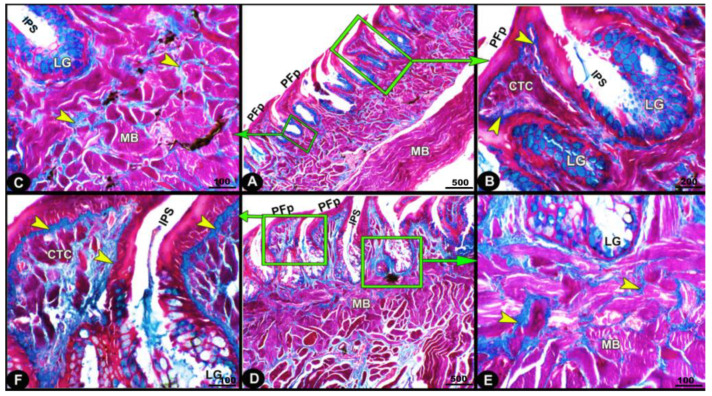
Micrograph image (views (**A**–**F**)) of the *Heremites vittatus* tongue showing the pointed filiform papillae (PFP), interpapillary space (IPS), connective tissue core (CTC), muscle bundles (MB), collagen fibers (yellow arrowheads) that were intertwined and filled the core of each papilla, and glands (LG). Masson’s trichrome stain.

**Figure 8 animals-13-03336-f008:**
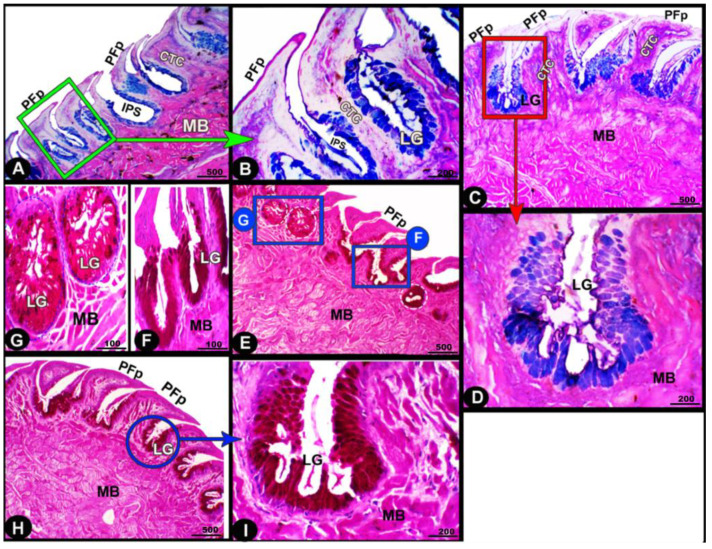
Histochemical micrograph image (views (**A**–**I**)) of the *Heremites vittatus* tongue showing the interpapillary space (IPS), the pointed filiform papillae (PFP), connective tissue core (CTC), muscle bundles (MB), and glands (LG). Note: the glands displayed strong AB and PAS-positive reactions, in which the blue color indicates positive AB reactivity while the red color indicates PAS reactions. PAS and AB stain.

**Figure 9 animals-13-03336-f009:**
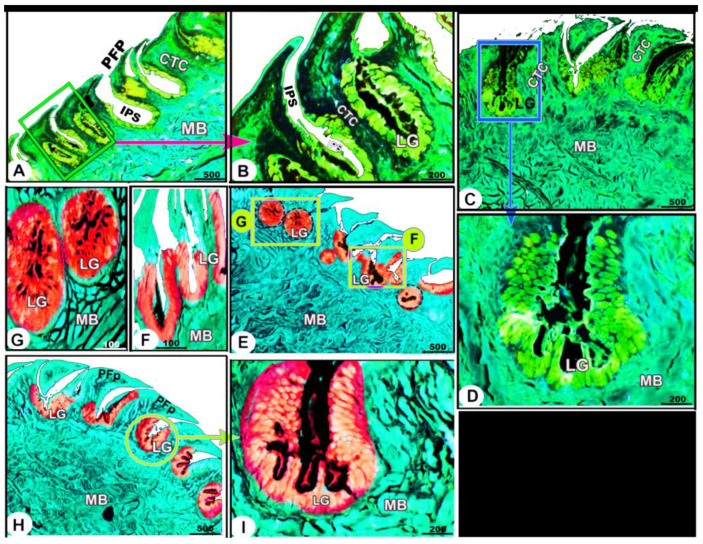
Histochemical micrograph image (views (**A**–**I**)) of the *Heremites vittatus* tongue showing the negative image of Figure 8 to clarify the pointed filiform papillae (PFP), interpapillary space (IPS), connective tissue core (CTC), muscle bundles (MB), and glands (LG). Note: the glands displayed strong AB and PAS-positive reactions, in which the blue color indicates positive AB reactivity while the red color indicates PAS reactions. PAS and AB stain.

**Figure 10 animals-13-03336-f010:**
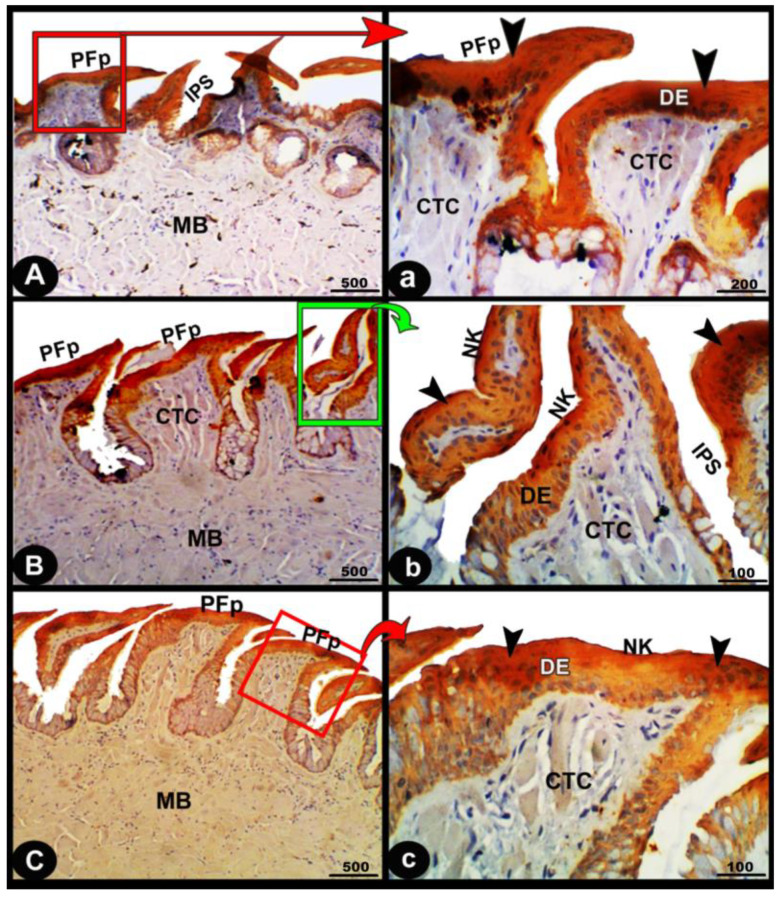
Immunohistological micrographs for cytokeratin of the dorsal surface of the foretongue (views (**A**,**a**)), midtongue (views (**B**,**b**)), and hindtongue (views (**C**,**c**)) of the *Heremites vittatus* tongue. The pointed filiform papillae (PFP), interpapillary space (IPS), connective tissue core (CTC), muscle bundles (MB), keratinized (KE) or non-keratinized (NK) squamous epithelium (DE), keratinized layer (black arrowheads), and glands (LG). Note: in immunohistochemistry analysis, there was a strong cytokeratin immunopositivity in all parts of the tongue (black arrowheads).

**Figure 11 animals-13-03336-f011:**
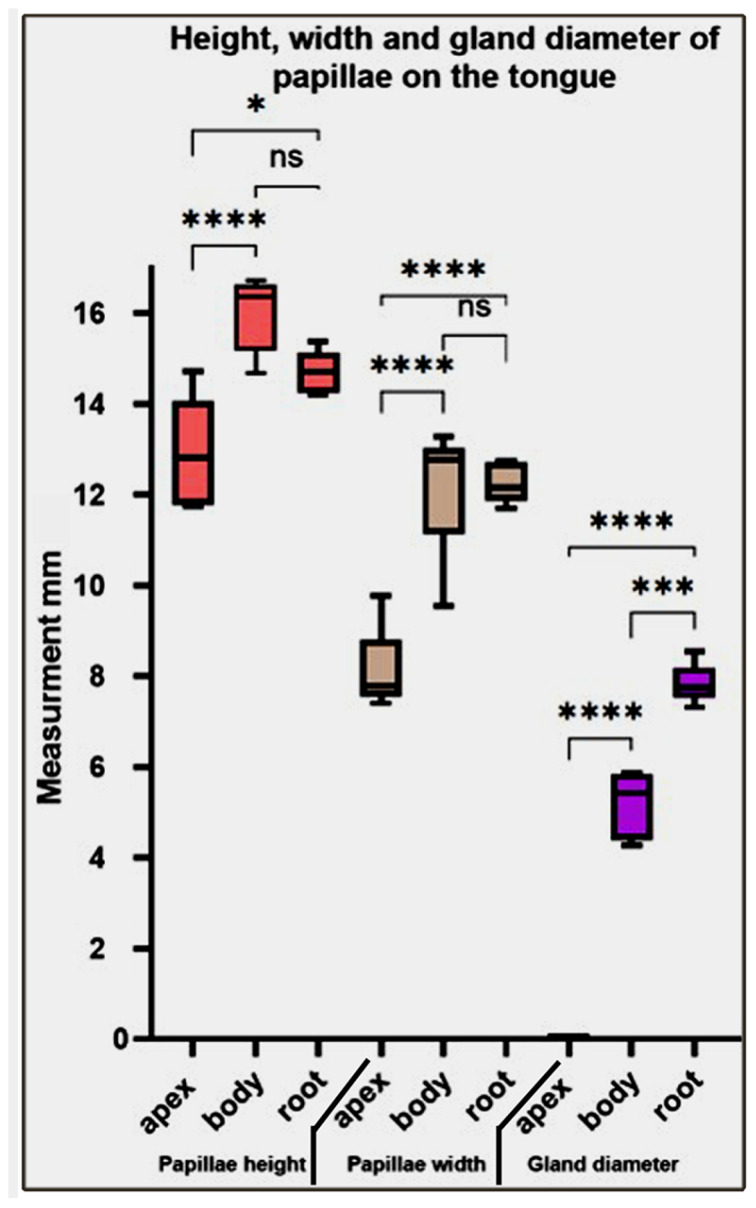
Represents the height, width, and gland diameter of the papillae on different parts of the *Heremites vittatus* tongue. Values have been represented as the mean ± SEM (*n* = 5). *, ***, and **** denote statistical significance with *p* < 0.05, *p* < 0.001, and *p* < 0.0001, respectively.

## Data Availability

The datasets used and/or analyzed during the current study are available from the corresponding author on reasonable request.

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
