# Peer review of "Tongue of the Egyptian Endemic Bridled Skink (Heremites vittatus; Olivier, 1804): Gross, Electron Microscopy, Histochemistry, and Immunohistochemical Analysis"

_animals, 2023, doi:10.3390/ani13213336_

Round 1
Reviewer 1 Report
Dear authors, you will find my comments directly in the manuscript.
It is about the right latin name, capturing the live wild animals for research and method of euthanasia, respectively. I am looking forward to read your manuscript after those revisions.

Author Response
Thank you for your valuable suggestions.
We have changed the right Latin name to Heremites vittatus in the revised version of the manuscript.
Reviewer 2 Report
The manuscript was presented rigorously. After review, we found no comments addressed to the co-authors. There is a need for proofreading to ensure that the scientific names are well written and to check the form of the text. On the other hand, we think that conclusion need to be improved. If we compare the importance of the discussions, it seems that the conclusion needs to be improved to balance the manuscript.
Congratulations to the authors.

Author Response
Thank you very much for your comments. We have revised our manuscript very carefully as per the comments of other reviewers and editors.
Thank you for your valuable suggestions. We have changed the right Latin name to Heremites vittatus in the revised version of the manuscript.
We adapted the conclusion section in relation to the discussion section according to your advice.
Reviewer 3 Report
The current study is unique and important to the field of ecology and wild species conservation. The photos obtained from the vast microscopy methods used (histology, histochemistry and SEM) are interesting and informative.
Yet, it suffers from some major problems, which require extensive re-editing.
· Lack of flow and coherence between paragraphs.
· Enormous disorder in the results section.
· Low English quality and lacked scientific writing.
Abstract:
· Over descriptive and mostly refer to the results section without addressing other sub-sections: aims, hypothesis, material and methods, discussion (the connection between the suggested conclusion and the results is not clear) – please revise.
· Shortcuts introduce for the first time, should be written also by their full names; for example: Ab – Alcian blue stain.
Introduction:
· There are major problems of coherency and flow; paragraphs seem to lack connection between them. I believe, one of the reasons for that, is the absent of connecting words between paragraphs.
· At the end of the introduction, clear hypothesis and research aims should be presented - Currently, they are absent.
· Lines 70-72: the sentence is not clear and need to be revised.
Material and methods:
· In general, section should be re-edited to ensure proper scientific writing and clarity regarding: procedures and reagents, which not always present; Reagents do not always have manufactures data and procedures are only partially described (for example: lack of incubation timing).
· Sample collection: in line 88, authors claim that "the reptiles were hunted and dissected" - what do you mean by hunted?
· Please add details regarding the animals' euthanasia, prior to the tissues samples collection.
· Please add details regarding the animals' husbandry in the university (water, light regime, temperature, humidity and density).
· AB and PAS stains, should be written in their full name, when introduce for the first time.
· According to the manufacture (Abcam), the anti pan-cytokeratin antibody used in the immunohistochemical analysis (ab80826) is reactive to humans – please provide any supportive data, that this antibody is also cross reactive to reptiles? Homology data on current investigated specie and human?
· Authors describe using photograph enhancement software for immunohistochemical analysis (lines 142-147). But that was done only for the negative control. This is not acceptable. Positive and negative images should be photographed in the same manner.
· Statistical analysis: it is not clear when t test and when tukey's tests were used – please clarify.
Results:
· Some of the figures suffer from visibility problems: low resolution, heading is too crowded and the text is partially superposed by the graph, Fonts are in different sizes in graphs, some do not have error bars… – this must be revised.
· I fail to understand what is the added value of comparing different segments size of the Trachylepis vittata? What is the biological value for performing statistical analysis for this type of data? Why not, just describe the findings in a table? – I think it would be much clearer.
· What is the meaning of Trachylepis vittata in the graph? – The whole animal?
· Results regarding figure 2 are described in the text per individual animal while in the figure; they are presented as a mean.
· Figure 3 contains panel of interesting photos which are only partially described in the text.
· Figure 8 appears before figure 7...
· There is an enormous disorder in the results section which makes it hard to understand which figure belongs to which text and legend.
Discussion and conclusions:
· Lines 415-417 are not clear –please revise.
· Lack of flow and coherence between paragraphs –please revise.
English and editing:
1. Why are the references and figure references in bold? – Please use same text font.
2. Please provide proofreading; grammar mistakes are abundant. For example: line 53: "a species", should be species.
3. Some of paragraphs contain sentences in different tenses (past and present) – this should be revised.
4. Morphological and morphometric examanations: the usage of the word "we" in this subsection is enormous and as of so impairs coherency and flow – please revise.
5. Line 167 is not aligned with the text – please fix.
Author Response
The current study is unique and important to the field of ecology and wild species conservation. The photos obtained from the vast microscopy methods used (histology, histochemistry and SEM) are interesting and informative.
Yet, it suffers from some major problems, which require extensive re-editing.
Lack of flow and coherence between paragraphs.
Enormous disorder in the results section.
Low English quality and lacked scientific writing.
Response:
Thank you for your valuable suggestions.
We have changed the right Latin name to Heremites vittatus in the revised version of the manuscript.
We adapted English grammar and sentence structure to be easy to understand by the readers according to your advice.
we adapted the flow and coherence between paragraphs according to your advice.
We adapted mous disorder in our results section according to your advice.
we adapted the English quality and scientific writing according to your advice.
Abstract:
Over descriptive and mostly refer to the results section without addressing other sub-sections: aims, hypothesis, material and methods, discussion (the connection between the suggested conclusion and the results is not clear) – please revise.
Shortcuts introduce for the first time, should be written also by their full names; for example: Ab – Alcian blue stain.
Response:
Thank you for your valuable suggestions.
We have adapted our abstract section by addressing to other subsections [aims, hypothesis, material and methods, discussion (the connection between the suggested conclusion and the results is not clear)] in the revised version of the manuscript according to your advice.
According to your advice, we have adapted the abbreviations by writing the complete names in the first time in the revised version of the manuscript.
Introduction:
There are major problems of coherency and flow; paragraphs seem to lack connection between them. I believe, one of the reasons for that, is the absent of connecting words between paragraphs.
At the end of the introduction, clear hypothesis and research aims should be presented - Currently, they are absent.
Lines 70-72: the sentence is not clear and need to be revised.
Response:
Thank you for your valuable suggestions.
According to your advice, we have adapted our introduction section by adapting the flow and coherence between paragraphs.
We adapted the connection between the paragraphs and rewrote the Line 70-72.
We rewrote and adapted the last paragraph of the introduction section by addressing our study's hypothesis and aims according to your advice.
Material and methods:
In general, section should be re-edited to ensure proper scientific writing and clarity regarding: procedures and reagents, which not always present; Reagents do not always have manufactures data and procedures are only partially described (for example: lack of incubation timing).
Sample collection: in line 88, authors claim that "the reptiles were hunted and dissected" - what do you mean by hunted?
Please add details regarding the animals' euthanasia, prior to the tissues samples collection.
Please add details regarding the animals' husbandry in the university (water, light regime, temperature, humidity and density).
AB and PAS stains, should be written in their full name, when introduce for the first time.
According to the manufacture (Abcam), the anti pan-cytokeratin antibody used in the immunohistochemical analysis (ab80826) is reactive to humans – please provide any supportive data, that this antibody is also cross reactive to reptiles? Homology data on current investigated specie and human?
Authors describe using photograph enhancement software for immunohistochemical analysis (lines 142-147). But that was done only for the negative control. This is not acceptable. Positive and negative images should be photographed in the same manner.
Statistical analysis: it is not clear when t test and when tukey's tests were used – please clarify.
Response:
Thank you for your valuable suggestions.
According to your advice, we scientifically rewrote our section and removed the aimless parts.
In line 88, we rewrote and adapted this paragraph according to your advice.
Before the tissue samples collection, we added a paragraph describing the animals' euthanasia.
According to your advice, we adapted this part animal housing mangment in the material and methods.
We rewrote the full name of AB and PAS stains when written firstly at the abstract section.
Kandyel, R., Elwan, M., Abumandour, M.M., El Nahass, E.E., 2021. Comparative ultrastructural‐functional characterizations of the skin in three reptile species; Chalcides ocellatus, Uromastyx aegyptia aegyptia, and Psammophis schokari aegyptia (FORSKAL, 1775): Adaptive strategies to their habitat. Microscopy Research and Technique 84, 2104–2118.
Statistical analysis:
Results:
Some of the figures suffer from visibility problems: low resolution, heading is too crowded and the text is partially superposed by the graph, Fonts are in different sizes in graphs, some do not have error bars… – this must be revised.
I fail to understand what is the added value of comparing different segments size of the Trachylepis vittata? What is the biological value for performing statistical analysis for this type of data? Why not, just describe the findings in a table? – I think it would be much clearer.
What is the meaning of Trachylepis vittata in the graph? – The whole animal?
Results regarding figure 2 are described in the text per individual animal while in the figure; they are presented as a mean.
Figure 3 contains panel of interesting photos which are only partially described in the text.
Figure 8 appears before figure 7...
There is an enormous disorder in the results section which makes it hard to understand which figure belongs to which text and legend.
Response:
Thank you for your valuable suggestions.
We adopted the resolution, prevented the overcrowded heading, and adapted the partially superposed text by the graph,
We adopted the fonts in the graphs and added the bars according to your advice.
We adapted and rewrote our section and removed the aimless parts.
We mean the whole name of Trachylepis vittata in the grap.
We adapted the arrangement of Figure 8 before Figure 7.
We adapted the arrangement of Figure 8 before Figure 7.
In our results section, we adapted and rewrote our results to prevent this disorder to easily understand which figure belongs to which text and legend.
Statistical analysis: In some of the images, we combined the length with the width, so we wanted to unify the use of measurement on the longitudinal axis of all figures.
Discussion and conclusions:
Lines 415-417 are not clear –please revise.
Lack of flow and coherence between paragraphs –please revise.
Comments on the Quality of English Language
Response:
Thank you for your valuable suggestions.
According to your advice, we have adapted our discussion and conclusion sections by adapting the flow and coherence between paragraphs.
We adapted the language quality of our article.
English and editing:
- Why are the references and figure references in bold? – Please use same text font.
- Please provide proofreading; grammar mistakes are abundant. For example: line 53: "a species", should be species.
- Some of paragraphs contain sentences in different tenses (past and present) – this should be revised.
- Morphological and morphometric examanations: the usage of the word "we" in this subsection is enormous and as of so impairs coherency and flow – please revise.
- Line 167 is not aligned with the text – please fix.
Response:
Thank you for your valuable suggestions.
We adapted the references section by removing the bold according to your advice.
We adapted the grammar mistakes of our article and corrected them very carefully in the revised version of the manuscript.
We adapted the tense of our sections in the revised version of the manuscript.
We remove the word "we" in this subsection (Morphological and morphometric examinations)
We have adapted the flow and coherence between paragraphs.
Round 2
Reviewer 3 Report
Abstract:
Issues were fixed
Introduction:
Issues were fixed
Material and methods:
· In general, section should be re-edited to ensure proper scientific writing and clarity regarding: procedures and reagents, which not always present; Reagents do not always have manufactures data and procedures are only partially describe – this issue still exists (for example: 2% formaldehyde and other SEM reagents).
· Please add details regarding the animals' husbandry in the university (water, light regime, temperature, humidity and density) – this information is still missing.
· According to the manufacture (Abcam), the anti pan-cytokeratin antibody used in the immunohistochemical analysis (ab80826) is reactive to humans – please provide any supportive data, that this antibody is also cross reactive to reptiles? Homology data on current investigated specie and human? This question remained unanswered.
· Authors describe using photograph enhancement software for immunohistochemical analysis (lines 142-147). But that was done only for the negative control. This is not acceptable. Positive and negative images should be photographed in the same manner – this comment remained unanswered.
· Statistical analysis: it is not clear when t test and when tukey's tests were used – please clarify - this comment remained unanswered.
Results:
· Some of the figures suffer from visibility problems: low resolution, heading is too crowded and the text is partially superposed by the graph, Fonts are in different sizes in graphs, some do not have error bars… – although authors commented that they revised and fixed this issue – there are no differences (when comparing to previous version and this issue consists.
· I fail to understand what is the added value of comparing different segments size of the Trachylepis vittata? What is the biological value for performing statistical analysis for this type of data? Why not, just describe the findings in a table? – I think it would be much clearer. This issue remained unfixed.
· What is the meaning of Trachylepis vittata in the graph? – The whole animal? This issue remained unfixed.
· Lines 311-315, referred to figure 7 and should be relocated post line 281, instead of current location (post figure 8)
Discussion and conclusions:
· Lines 415-417 are not clear –although, the author commented these lines were revised, they are exactly the same as in version 1.
· Lack of flow and coherence between paragraphs – this issue was only partially solved. There is still a flow problem (in my opinion, the frequent repetitiveness of the connecting word "our" to start each paragraph at the last third section of the discussion, is the main cause for that) – this issue should be revised completely.
English and editing:
1. Some of the references, figures refrences are still in bold – Please use same text font.
2. Moderate English editing is still needed.
Author Response
Material and methods:
In general, section should be re-edited to ensure proper scientific writing and clarity regarding: procedures and reagents, which not always present; Reagents do not always have manufactures data and procedures are only partially describe – this issue still exists (for example: 2% formaldehyde and other SEM reagents).
Response: Thank you very much for your comments. We have revised our manuscript very carefully as per the comments of other reviewers and editor. Thank you for your valuable suggestions. We adapted and re-edited this section to ensure proper scientific writing and clarity regarding procedures and reagents. All the changes have been highlighted in the revised version of the manuscript.
Please add details regarding the animals' husbandry in the university (water, light regime, temperature, humidity and density) – this information is still missing.
Response: Thank you very much for your comments. We have revised our manuscript very carefully as per the comments of other reviewers and editors. Thank you for your valuable suggestions. We added a sentence describing the details regarding the animals' husbandry in the university (water, light regime, temperature, humidity and density) as the follows; (Housing includes a reptile's cage with a temperature range of 18–24 °C and 55–70% humidity, Crickets, mealworms, cockroaches, earthworms, superworms, hornworms, and other prey animals can all be found in diets). All the changes have been highlighted in the revised version of the manuscript.
According to the manufacture (Abcam), the anti pan-cytokeratin antibody used in the immunohistochemical analysis (ab80826) is reactive to humans – please provide any supportive data, that this antibody is also cross reactive to reptiles? Homology data on current investigated specie and human? This question remained unanswered.
Response: Thank you very much for your comments. We have revised our manuscript very carefully as per the comments of other reviewers and editor. Thank you for your valuable suggestions We adapted the portion related to the manufacture (Abcam), the anti pan-cytokeratin antibody used in the immunohistochemical analysis (ab80826) is reactive to humans a ccoridng to your advice in the material and methods section.
Authors describe using photograph enhancement software for immunohistochemical analysis (lines 142-147). But that was done only for the negative control. This is not acceptable. Positive and negative images should be photographed in the same manner – this comment remained unanswered.
Response: Thank you very much for your comments. We have revised our manuscript very carefully as per the comments of other reviewers and editor. Thank you for your valuable suggestions . We adapted this portion related to the using photograph enhancement software for immunohistochemical analysis (lines 142-147). All the changes have been highlighted in the revised version of the manuscript.
Statistical analysis: it is not clear when t test and when tukey's tests were used – please clarify - this comment remained unanswered.
Response: Thank you very much for your comments. We have revised our manuscript very carefully as per the comments of other reviewers and editor. Thank you for your valuable suggestions.
Statistical analysis: Tukey's method is used in ANOVA to create confidence intervals for all pairwise differences between factor level means while controlling the family error rate to a level you specify. The Tukey HSD ("honestly significant difference" or "honest significant difference") test is a statistical tool used to determine if the relationship between two sets of data is statistically significant – that is, whether there's a strong chance that an observed numerical change in one value is causally related to an observed change in another value. In other words, the Tukey test is a way to test an experimental hypothesis.
Results:
Some of the figures suffer from visibility problems: low resolution, heading is too crowded and the text is partially superposed by the graph, Fonts are in different sizes in graphs, some do not have error bars… – although authors commented that they revised and fixed this issue – there are no differences (when comparing to previous version and this issue consists.
Response: Thank you very much for your comments. We have revised our manuscript very carefully as per the comments of other reviewers and editor. Thank you for your valuable suggestions We adapted the figure resolution, remove the heading, and the fonts. we added the scale bars in the figures. All the changes have been highlighted in the revised version of the manuscript.
I fail to understand what is the added value of comparing different segments size of the Trachylepis vittata? What is the biological value for performing statistical analysis for this type of data? Why not, just describe the findings in a table? – I think it would be much clearer. This issue remained unfixed.
Response: Thank you very much for your comments. We have revised our manuscript very carefully as per the comments of other reviewers and editor. Thank you for your valuable suggestions. Statistical analysis: In some of the images, we combined the length with the width, so we wanted to unify the use of measurement on the longitudinal axis of all figures. We confirmed that this statistical comparison between the length of the lizard and its different parts, especially the tongue, will give a biological value to describe their adaptation in its environment if compared to other species.
What is the meaning of Trachylepis vittata in the graph? – The whole animal? This issue remained unfixed.
Response: Thank you for your valuable suggestions. We removed this from graph and adapted the figures.
Lines 311-315, referred to figure 7 and should be relocated post line 281, instead of current location (post figure 8)
Response: Thank you for your suggestion. We have relocated the suggested lines before figure 7 in the revised mansucript. The same has been highlighted in the revised version of the manuscript.
Discussion and conclusion
Lines 415-417 are not clear –although, the author commented these lines were revised, they are exactly the same as in version 1.
Response: Thank you for your valuable suggestions. As per your suggestions, we updated Lines 415–417 of the discussion section and added other sentences after this sentence to discuss what we needed by adding this sentence.
Lack of flow and coherence between paragraphs – this issue was only partially solved. There is still a flow problem (in my opinion, the frequent repetitiveness of the connecting word "our" to start each paragraph at the last third section of the discussion, is the main cause for that) – this issue should be revised completely.
Response: Thank you for your valuable suggestions. We adapted the problem of the the frequent repetitiveness of the connecting word "our" to start each paragraph at the last third section of the discussion, is the main cause for that).
Comments on the Quality of English Language: English and editing:
Some of the references, figures references are still in bold – Please use same text font.
Moderate English editing is still needed.
Response: Thank you for your valuable suggestions. We adapted English grammar and sentence structure to be easy to understand by the readers according to your advice. We adapted the flow and coherence between paragraphs according to your advice. We adapted the English quality and scientific writing according to your advice. We adapted the references and figures references by removing the bold according to your advice.